# Enhanced Electrochromic Properties by Improvement of Crystallinity for Sputtered WO$_3$ Film

**Zhu-jie Xia [1,2], Hong-li Wang [2,3], Yi-fan Su [2], Peng Tang [2], Ming-jiang Dai [2], Huai-jun Lin [1], Zhi-guo Zhang [1,*] and Qian Shi [2,3,*]**

[1] Institute of Advanced Wear and Corrosion Resistant and Functional Materials, Jinan University, Guangzhou 510632, China; xiazhujie@stu2017.jun.edu.cn (Z.-j.X.); hjlin@jnu.edu.cn (H.-j.L.)

[2] The Key Lab of Guangdong for Modern Surface Engineering, National Engineering Laboratory for Modern Materials Surface Engineering Technology, Guangdong Institute of New Materials, Guangdong Academy of Science, Guangzhou 510651, China; wanghongli@gdinm.com (H.-l.W.); suyifan@gdinm.com (Y.-f.S.); tangpeng@gdinm.com (P.T.); daimingjiang@tsinghua.org.cn (M.-j.D.)

[3] Guangdong Provincial Key Laboratory of Advanced Energy Storage Materials, South China University of Technology, Guangzhou 510641, China

\* Correspondence: zhigzhang@jnu.edu.cn (Z.-g.Z.); qianzixlf@163.com (Q.S.)

**Abstract:** Tungsten oxide (WO$_3$) is widely used as a functional material for "smart windows" due to its excellent electrochromic properties, however it is difficult to overcome the conflict between its optical modulation and cyclic stability. In this work, WO$_3$ thin films with different crystal structures were prepared by DC reactive magnetron sputtering method. The effects of substrate temperature on the structure, composition, and electrochromic properties of WO$_3$ films were investigated. The results show that the crystallinity of the WO$_3$ film increases with increasing deposition temperature, indicating that temperature plays an important role in controlling the structure of the WO$_3$ film. For WO$_3$ thin films formed at a substrate temperature of 573 K, the film is in an amorphous state to a crystalline transition state. From X-ray diffraction (XRD) analysis, the thin film showed a weak WO$_3$ crystallization peak, which was in the composite structure of amorphous and nanocrystalline. Which has the best electrochromic properties, with modulation amplitude of 73.1% and bleached state with a coloration efficiency of 42.9 cm$^2$/C at a wavelength of 550 nm. Even after 1500 cycles, the optical modulation still contains 65.4%, delivering the best cyclic stability.

**Keywords:** DC reactive magnetron sputtering; WO$_3$ thin film; crystallinity; electrochromic; substrate temperature

## 1. Introduction

With the increase of global challenges related to energy depletion, the development of new energy-saving materials, such as electrochromic (EC) materials attracting people's attention [1–3]. Electrochromic materials are capable of reversibly changing their optical properties by switching the applied voltage [4–6]. Owing to their large optical modulation range and long cycle life, the EC materials can be used as a smart window for energy-saving buildings or antiglare rear-view mirrors for automobiles. Tungsten trioxide is considered to be a promising EC material, which achieves high optical modulation from transparency to blue by lithium ion and electron injection/extraction [7–11].

Magnetron sputtering is the most promising technology for WO$_3$ thin film preparation. Studies have shown that the electrochromic performance of WO$_3$ thin films is closely related to its structure, including geometry, particle size, crystallinity, stoichiometry, etc. [12,13]. It is widely believed that crystalline WO$_3$ thin films have remarkable cyclic stability and low optical modulation range, while

the amorphous ones show the opposite characteristics [14,15]. Researchers have been devoted to balancing the different EC properties between amorphous and crystalline $WO_3$ films [16]. For example, Gaurav M. et al. [17] prepared nanoporous γ-$WO_3$ thin films by DC reactive magnetron sputtering and subsequent heat treatment. The results show that its light modulation is 46%, good cyclic stability at least up to 500 cycles. However, the results are still unsatisfactory because of nonindustrial post annealing process, in situ preparation and poor EC performance. Related researchers are concerned about how to improve the crystalline characteristics of the film to obtain high EC performance. Such as, Madhavi V. et al. [18] tried to improve the crystallinity of $WO_3$ films by adjusting substrate temperature during RF reactive magnetron sputtering process, however the poor optical modulation of 40% and coloration efficiency of 33.8 $cm^2$/C were obtained.

The magnetron sputtering method has been widely used to prepare oxide films with adjustable microstructure and crystalline structure. In this study, $WO_3$ thin films with intermediate states between amorphous and nanocrystalline were prepared by DC reactive magnetron sputtering [19]. The structure, composition, and electrochromic properties of $WO_3$ films with different crystallinity were obtained and their impact on performance was discussed. Based on the different substrate temperature during deposition, we are trying to reveal the relationship between the crystal level and EC properties.

## 2. Materials and Methods

Tungsten trioxide thin films were grown on $In_2O_3$:Sn (ITO film sheet resistance of about 8 Ω) coated glass substrates by DC reactive magnetron sputtering technique using pure metallic tungsten target (76.2 mm diameter and 3 mm thickness). The base pressure is pumped to below $8.0 \times 10^{-4}$ Pa, then the $O_2$ and Ar (99.99% purity) gas with a ratio of 1:1 are applied as reaction gas and sputtering gas. During deposition, the sputtering power and working pressure were kept at 270 W and 3 Pa respectively. The target–substrate distance was 76.5 mm. To obtain the different crystallinity of $WO_3$ thin films, the films were prepared at substrate temperatures of 300, 473, 573, 673 and 743 K, respectively.

The crystallographic structure of films was analyzed by X-ray diffraction (XRD, D8 Advance X, Bruker, Karlsruhe, Germany), using Cu-Kα as target with scanning range of 20–80° and step size of 0.05°. The film composition, bonding energy and bonding state of tungsten and oxygen were measured by X-ray photoelectron spectroscopy (XPS, AXIS Supra, Shimadzu, Tokyo, Japan). The microstructure and morphology of the films were characterized by field emission scanning electron microscope (FESEM, SU8220, Hitachi, Tokyo, Japan) and atomic force microscope (AFM, Dimension edge, Bruker, Karlsruhe, Germany). The transmittance of the thin film was measured by an ultraviolet-visible spectrophotometer (SP-752PC, Spectrum, Shanghai, China) with a wavelength range of 300–800 nm. The electrochromic and electrochemical performance were tested by electrochemical workstation (P4000, Princeton, NJ, USA).

## 3. Results and Discussion

### 3.1. Structure and Composition

XRD was employed to determine the crystal structure and possible phase changes of $WO_3$ thin films on different substrate temperatures, as the results are shown in Figure 1. It can be seen that there is no diffraction peaks for $WO_3$ films at low substrate temperature of 300–473 K except for the peaks of ITO glass. The low intense peaks near 24° and 34° belonging to $WO_3$ are observed at substrate temperature of 573 K, revealing the crystallinity transition. When the substrate temperature rises above 673 K, the positions of the reflection planes of $WO_3$ film are 2θ = 23.1° (002), 2θ = 23.5° (020), and 2θ = 24.1° (200) corresponding to the monoclinic structure [JCPDS. 43-1035]. It can also be seen that the intensity of the Bragg reflection increases with the substrate temperature, indicating that the film has higher degree of crystallization at higher substrate temperature [20]. The results reveal that the crystal level of the $WO_3$ can be controlled. The films deposited at 473 K are non-crystalline with amorphous properties. When increasing the temperature to 673 K, the (002), (020), and (200) diffraction appear

indicating that the film grows into complete monoclinic phase crystals. Meanwhile, Scherrer formula was used to evaluate the microcrystalline size of the $WO_3$ film, and the grain size was calculated to be approximately 12–28 nm. For the film deposited at 573 K, the wide and weak diffraction peaks at (002) and (400) indicate that the film is in a transition state between amorphous and nanocrystalline states [21].

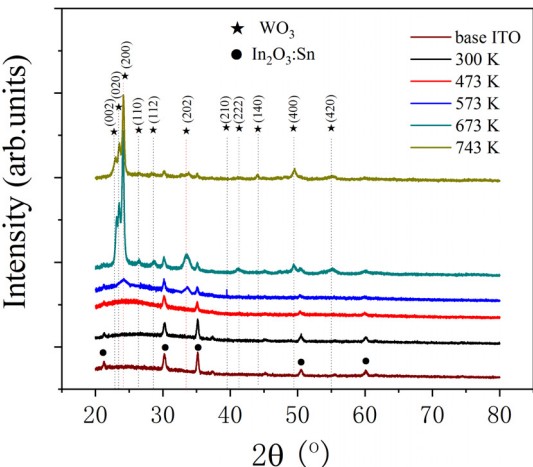

**Figure 1.** X-ray diffraction (XRD) patterns of the $WO_3$ thin films deposited at different substrate temperature.

XPS analysis was performed to investigate the chemical composition and reveal the elemental binding states of $WO_3$ films. Figure 2a shows the full-scan XPS spectrum of $WO_3$ films, mainly exhibiting W and O. It is calculated from the XPS element peak intensity that the atomic ratio W/O decreases with increasing substrate temperature may be related to the oxidation state of tungsten, because substrate temperature will affect the degree of reaction between the tungsten atoms and oxygen and influences the W and O combination of the film. Figure 2b shows the W4*f* spin-orbit energy level spectrum as a function of substrate temperature and the Gaussian fitted curve for film obtained at 573 K. There are continuous three spectrum corresponding to W4$f_{7/2}$, W4$f_{5/2}$, and W4$f_{3/2}$ peaks which are located at 35.4, 37.7, and 41.3 eV, respectively, matching well with those reported in the literature for $WO_3$ film [22]. The peaks at 35.4 and 37.7 eV were attributed to W4$f_{7/2}$ and W4$f_{5/2}$ for W(VI), whereas the peaks at 34.9 and 37.1 eV were attributed to W4$f_{7/2}$ and W4$f_{5/2}$ for W(V). Moreover, W4$f_{7/2}$ and W4$f_{5/2}$ can be split into W(VI) with higher binding energy and W(V) double peak with lower binding energy. However, the electrochromic process of $WO_3$ film mainly realizes the mutual conversion between W(VI) and W(V), and absorbs the photon energy to undergo color conversion. Figure 2c shows that the peak position of the W4*f* spin trajectory shifts to the direction of low binding energy, and peak intensity also decreases as the substrate temperature increases. As a result, the ratio of W(VI)/W(V) in the obtained film is reduced, which restricts the process of film electrochromic from W(VI) to W(V), which affects its corresponding electrochromic performance. Figure 2d gives O1*s* spin orbit energy spectrum of the film at different substrate temperatures. The continuous double peaks represent the lattice oxygen ($\alpha$) and the interstitial oxygen ($\beta$), respectively, corresponding to the binding energy of 530.7 and 532.0 eV. The relative content of the oxygen vacancy for $WO_3$ films was evaluated according to the area ratio of these two integral components: ($\beta$ peak)/($\alpha$ peak + $\beta$ peak), as the substrate temperature increased from 300 to 743 K, the surface oxygen vacancy content was 14.8%, 17.0%, 20.0%, 34.1% and 34.4%. As the substrate temperature increases oxygen vacancy concentration increases, indicating that the binding energy of the film is relatively low. In addition, the O1*s* binding energy also moved to a lower binding energy position, which accompany with phase transition of the film changed from amorphous to crystalline [23]. It can be known that as the substrate temperature increases, the peaks of W4*f* and O1*s* spin orbits move toward the lower binding energy, indicating lower chemical bond and more stable structure at higher substrate temperatures. The results show

that the crystalline state has lower binding energy and more oxygen vacancy than the amorphous tungsten oxide.

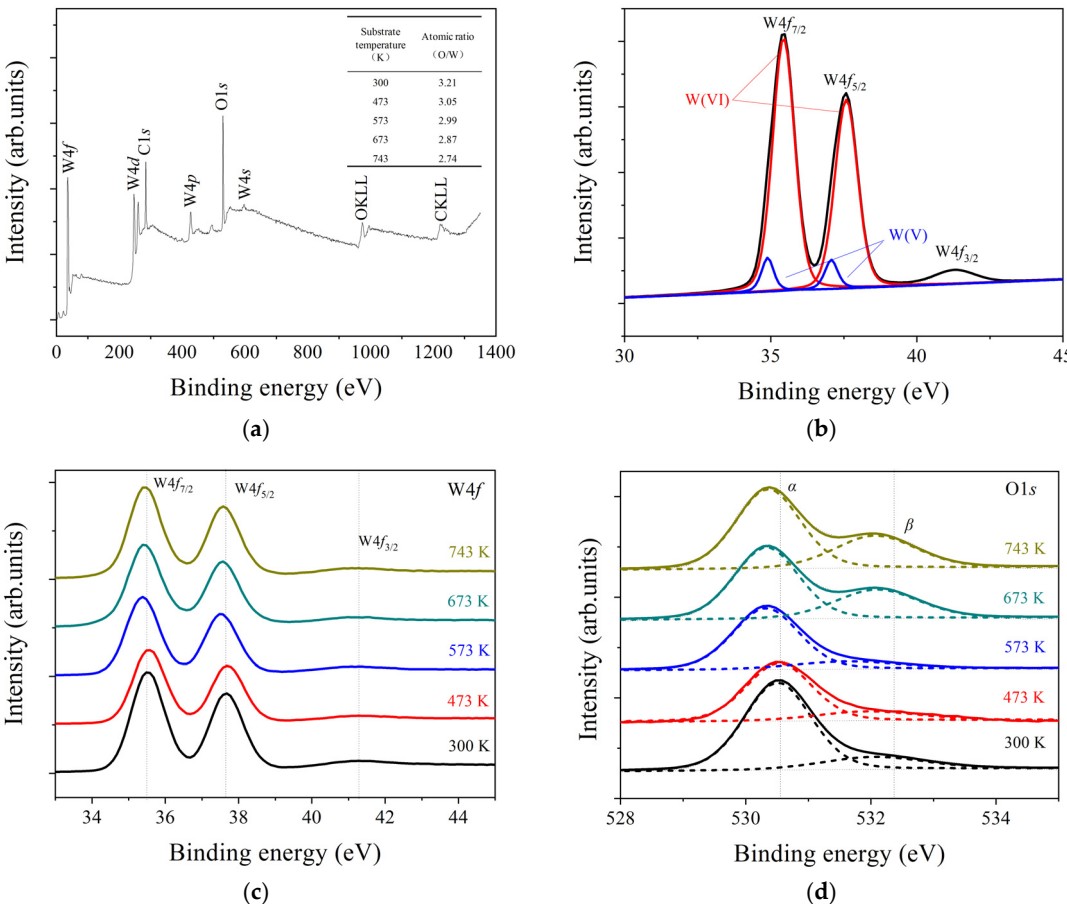

**Figure 2.** The WO$_3$ thin films deposited at different substrate temperatures: (**a**) X-ray photoelectron spectroscopy (XPS) full spectra, (**b**) W4*f* Gaussian fitting peak spectra, (**c**) W4*f* spin orbit energy level spectra and (**d**) O1*s* spin orbit energy level spectra.

### 3.2. Microscopic and Morphology

The prepared sample is an ITO and WO$_3$ double layer film, in which the ITO as conductive layer and certain thickness of WO$_3$ film is plated on the surface layer. The surface and cross-sectional SEM images of WO$_3$ thin films at different substrate temperatures are illustrated in Figure 3. The literature reports that substrate temperature has a great influence on the growth of WO$_3$ film [24]. When substrate temperature is 300 K, the film presents compact and smooth surface (Figure 3a). For the WO$_3$ film prepared at 473 and 573 K, there is still a uniformly distributed spherical grain structure (Figure 3b,c). As the substrate temperature rises to 673 K, the spherical particles become islands and flakes (Figure 3d,e). At the same time, the crystalline state of the thin film changes, and the particle size of the thin film becomes small and flakes pile up. It also can be seen from the cross-sectional view that as the substrate temperature increases, the thin film growth leads to higher density. And the growth of the columnar crystal of the crystalline film becomes more obvious, which corresponds to the crystal orientation in the XRD pattern (200).

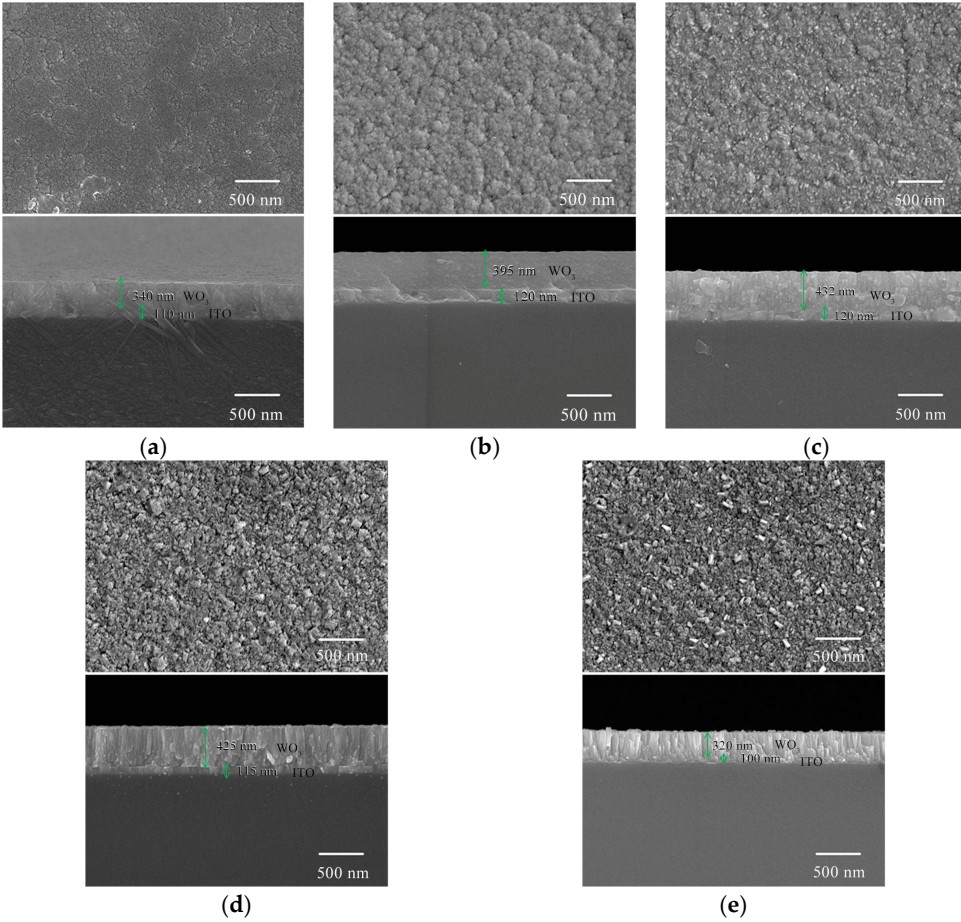

**Figure 3.** Surface and cross section scanning electron microscope (SEM) images of WO$_3$ thin films deposited at different substrate temperatures (**a**) 300 K, (**b**) 473 K, (**c**) 573 K, (**d**) 673 K and (**e**) 743 K.

Furthermore, the surface roughness of WO$_3$ film was characterized by AFM to analyze the film surface patterns grown at different substrate temperatures. The mean surface roughness (Rms) increases as a function of the substrate temperature as shown in Figure 4f. Due to the fact that as the substrate temperature increases, the deposited particles can obtain sufficient energy to migrate along the surface and inside, and combine with other particles to form crystals, and the crystal growth rate accelerates and the crystal grains become larger, resulting in an increase in surface roughness [25]. Figure 4a shows that the surface of the film is peak-shaped, and the roughness of amorphous WO$_3$ thin film is approximately 2.43 nm. However, as the substrate temperature increases, the surface roughness becomes larger and the particle size becomes smaller. Since the electrochemical reaction first occurs on the surface of the film, the morphology may show important influence on the implantation and extraction of lithium ions, and therefore the effects observed in the current changes associated with the oxidation and reduction processes will be further discussed.

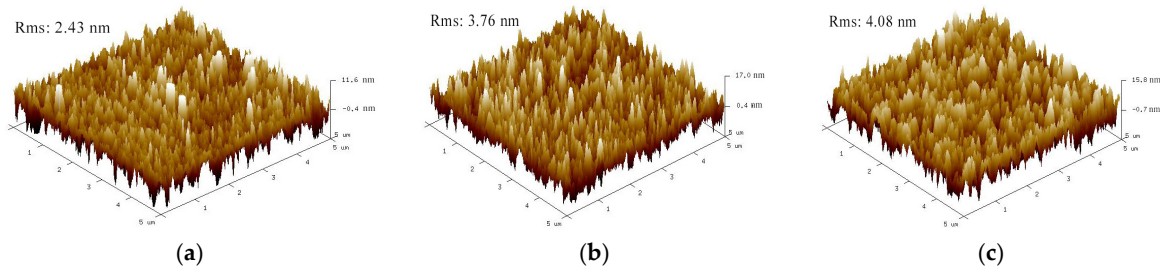

**Figure 4.** *Cont.*

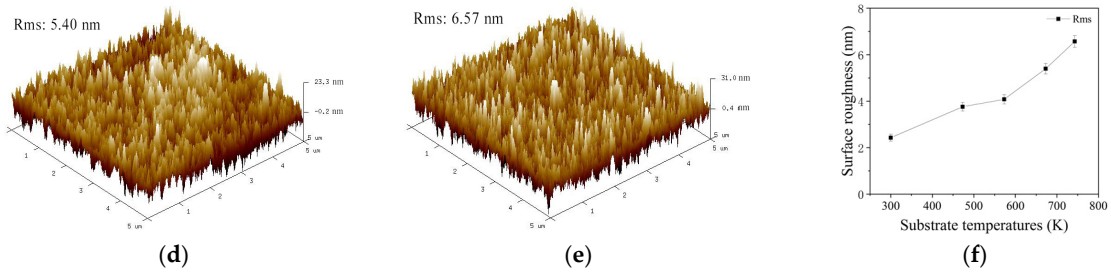

**Figure 4.** Surface atomic force microscope (AFM) images of WO₃ thin films deposited at different substrate temperatures (**a**) 300 K, (**b**) 473 K, (**c**) 573 K, (**d**) 673 K, (**e**) 743 K and (**f**) R_a curve.

### 3.3. Optical Properties

Figure 5a shows the optical transmittance and absorbance of the WO₃ films at different substrate temperatures in wavelength range of 300–800 nm. When comparing the transmittance of films at different substrate temperatures, the test deducts the ITO backing. The experiment keeps the thickness of the WO₃ film within a certain range of 320–432nm, experimental tests confirm that the thickness has little effect on transmittance. The transmittance curve shows that as the substrate temperature increases, the interference wavelength of the transmittance of the WO₃ film becomes larger, which means that the absorption coefficient of the film becomes smaller. The decrease in transmittance at higher substrate temperatures may be due to increase in surface roughness and improved crystallinity of the film resulting in more loss of light scattering. Generally, the sharp drop of transmittance was due to the fundamental absorption of light caused by the excitation of electrons from the valance band to the conduction band. It can be seen from the light absorption curve that the absorption edge moves to lower wavelength region as the substrate temperature increases, which called "blue shift" phenomenon [26]. It can be attributed to the presence of W(V) in the crystalline structure creating mid-gap defect states. Figure 5b shows the relationship between $(\alpha h v)^2$ and photon energy of the films formed at different substrate temperatures. When the substrate temperature is 473 K, the maximum optical band gap ($E_g$) of the film is 3.75 eV. The increased band gap can be attributed to more oxygen vacancy and enhanced crystallization of the film, making the refractive index of the film to light increase [27]. This is because it refracts light and increases the band gap of the film, high substrate temperature enhances regular arrangement of thin film, and the accompanying increased oxygen vacancy and crystalline regular structure will produce a high refractive index and absorption coefficient becomes smaller, resulting in a smaller optical band gap. Therefore, the width of the forbidden band increases with the substrate temperature, resulting in weakening the optical performance of the prepared WO₃ film.

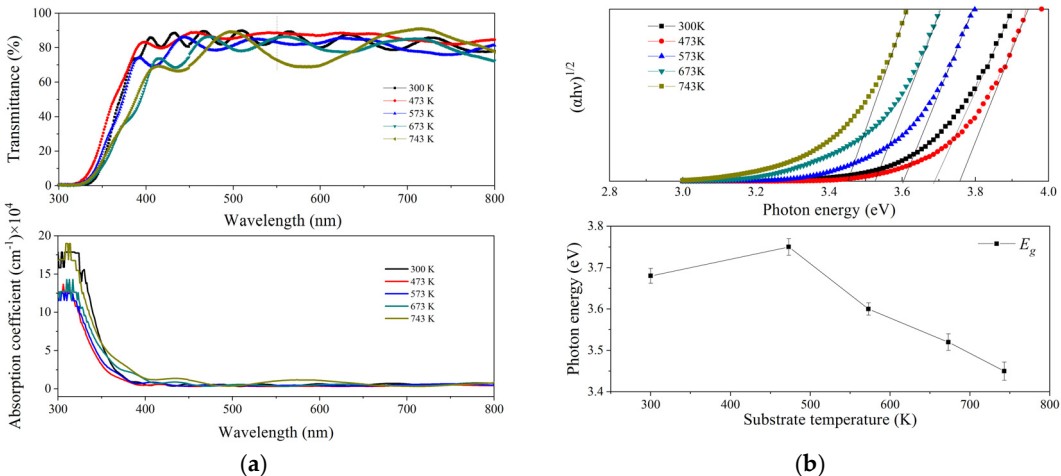

**Figure 5.** The WO₃ films deposited at different substrate temperature: (**a**) transmittance and absorption coefficient curve and (**b**) Tuac plot and $E_g$ curve.

### 3.4. Electrochromic and Electrochemical Characterization

In order to investigate the electrochromic behavior of the films, the integrated area of the cyclic voltammetry curve and the position of the anode peak are calculated and identified, since they are closely related to the ion insertion/extraction and electrochemical process occurring in the electrochromic films. The principle of electrochromism is generally recognized by the double injection model of ions and electrons [28]. This process can be represented by the following reversible electrochemical reaction:

$$WO_3 \text{ (transparent)} + Li^+ + e^- \boxed{\leftrightarrow} Li_xWO_3 \text{ (deep blue)} \qquad (1)$$

Typical cyclic voltammogram (CV) and lithium ion ($Li^+$) diffusion efficiency for the $WO_3$ film at different substrate temperatures are shown in Figure 6. It is considered that the larger integrated area of the CV curve, the more charge injected into the film, and the high anode peak position means greater $Li^+$ diffusion efficiency. The $Li^+$ diffusion coefficient ($D$) was determined according to the Randles–Sevcik equation [29]:

$$I_p = 2.72 \times 10^5 \times D^{(1/2)} \times C_0 \times v^{(1/2)} \qquad (2)$$

where $I_p$ is the anodic peak current density, $C_0$ the concentration of $Li^+$ in the electrolyte, and $v$ the scan rate for CV test. As shown in Figure 6, when substrate temperature increases from 300 to 473 K the $I_p$ increases, indicating the improved electrochemical characteristics. For the film prepared at 573 K, the CV integral curve area is the largest with the maximum anodic peak current density of 2.02 mA/cm$^2$. After that, $I_p$ sharply decreases. The calculated $Li^+$ diffusion efficiency is shown in the inset of the Figure 6. According to the Randles–Sevcik equation, the maximum $Li^+$ diffusion coefficient obtained at 573 K is $4.40 \times 10^{-11}$ cm$^2$/s, it is superior to the previous Madhavi V. literature reports that the best diffusion efficiency of the prepared film ion is $1.85 \times 10^{-11}$ cm$^2$/s. It can be attributed to the increased surface roughness and increased micro-void concentration of the film. When the substrate temperature is appropriately increased to 573 K, the loss structures between the amorphous and crystalline states can provide more effective contact area. The enhanced surface ion-implanting activity leads to an increase in the diffusion coefficient, which is in favor of lithium ion insertion/extraction [30]. However, the further substrate temperature increasing enhances the crystallinity and the formation of columnar crystal, which may cause a regular decrease in the $Li^+$ diffusion coefficient.

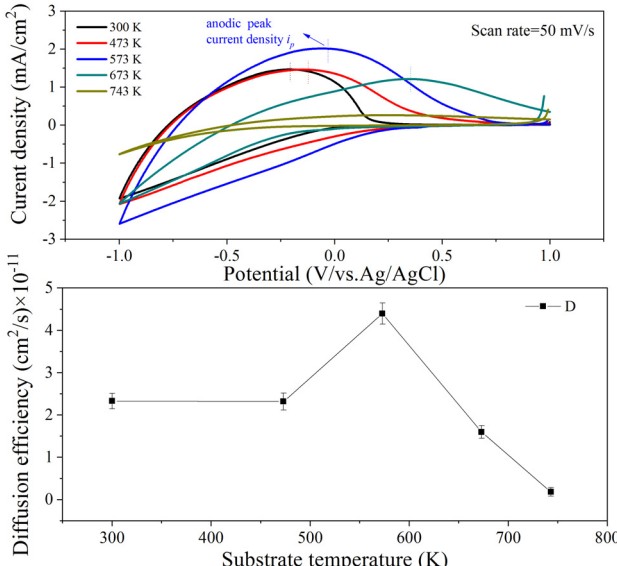

**Figure 6.** Cyclic voltammetry (CV) and diffusion efficiency curve of $WO_3$ films deposited at different substrate temperatures.

The response time which is obtained by the trend of the current density change is a main evaluation indexes for electrochromic performance. The electrochemical test is performed by applying alternative ±1 V constant potentials, Figure 7 shows constant current step current density and charge density as a function of time. The calculated coloring time ($t_c$) and bleaching time ($t_b$) values are given in Table 1. When the substrate temperature is 300 K, the peak value of the response of tungsten oxide is 6.8 mA/cm$^2$ and the maximum injection charge is −30.2 mC/cm$^2$. For film prepared at higher substrate temperature, large charge density and a short fading time were detected. The maximum response current peak density obtained at 573 K is 15.3 mA/cm$^2$, and the injected charge density is as high as −46.4 mC/cm$^2$. After that, the degraded electrochromic properties with low response current density, small injection amount and long fading time were received at substrate temperature of 573–743 K. This trend is more pronounced with increases temperature, which should be attributed to their different microstructures especially the crystallinity. The WO$_3$ films with amorphous structure provide more lithium ion diffusion channels and storage space, however the regularity and compaction crystal structure will weaken their corresponding electrochemical characteristics [31,32]. Figure 7 shows that the film obtained at substrate temperature of 573 K has a maximum substrate response current density and injected charge density, which is consistent with the CV results.

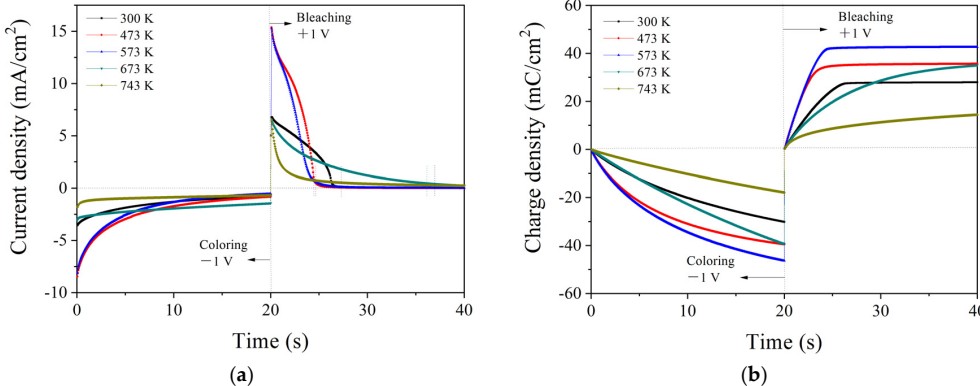

**Figure 7.** The WO$_3$ films deposited at different substrate temperatures: (**a**) chronoamperometry (CA) and (**b**) chronocoulometry (CC) curves.

**Table 1.** Electrochromic index various parameter of WO$_3$ films deposited at different substrate temperatures.

| Substrate Temperatures (K) | Optical Modulation Range (%) | | Response Time (s) | | Coloration Efficiency (cm$^2$/C) | |
|---|---|---|---|---|---|---|
| | 550 nm | 630 nm | $t_b$ | $t_c$ | 550 nm | 630 nm |
| 300 | 70.6 | 75.3 | 6.3 | 13.5 | 32.9 | 47.4 |
| 473 | 72.1 | 78.4 | 4.4 | 11.0 | 37.9 | 52.4 |
| 573 | 73.1 | 79.9 | 3.9 | 10.5 | 42.9 | 60.3 |
| 673 | 60.8 | 67.7 | 14.4 | 15.5 | 25.9 | 43.2 |
| 743 | 44.0 | 56.6 | 16.4 | 17.2 | 19.8 | 38.3 |

The detailed parameters for the electrochromic properties of tungsten oxide films prepared at different substrate temperatures are given in Table 1. The coloring efficiency (CE) in the wavelength range of 300–800 nm demonstrates the same regularity as the previously tested cyclic voltammetry (CV) and chronoamperometry (CA). As the temperature increases, the coloring efficiency first increases and then decreases. When the substrate temperature is 573 K, at 550 and 630 nm Wavelengths, the coloring efficiencies are maximum at 42.9 and 60.3 cm$^2$/C, respectively. It is noticed that the decrease in coloring efficiency is accompanied with the increase of crystallinity degree. It is reported crystalline WO$_3$ with O–W–O network interacts with lithium ions and forms a Li$_x$WO$_3$ phase, which may reduce the coloration efficiency [33,34].

The durability is important for practical applications of WO$_3$ films. It is known that crystal structures have important significance for the stability of electrochromic properties of WO$_3$ thin films [35,36]. The above results show that the between amorphous and nanocrystalline state WO$_3$ film with a substrate temperature of 573 K has a large modulation amplitude, a fast response speed, and a high coloring efficiency. It is necessary to analyze the cyclic stability of WO$_3$ film under different substrate temperatures. Figure 8 shows the transmittance variation for the WO$_3$ thin film after 1500 cycles at different substrate temperatures, there is obvious cyclic degradation for the film with amorphous structure modulation amplitude retention rate of 89.5% after 1500 cycles, indicating the excellent cycle stability at substrate temperature under 573 K, however crystalline films can provide stable cycling performance. This is due to the low binding energy of the crystalline WO$_3$ film, which is more regular and stable, and has a distinct columnar structure and crystal orientation. During the WO$_3$ thin film is periodically colored process, the columnar and orientation crystal can provide shorter migration distance for lithium ions [37,38]. It is noticed that although the attenuation of the amorphous WO$_3$ film is even worse after 1500 cycles, it still has better color rendering ability than the crystal one. In general, for the film deposited at 573 K has the best electrochromic properties including the shortest response time, the largest optical modulation and the most stability. This is because the film with combined nanocrystalline and amorphous structure can provide more sites and short diffusion path for lithium ion injection/extraction.

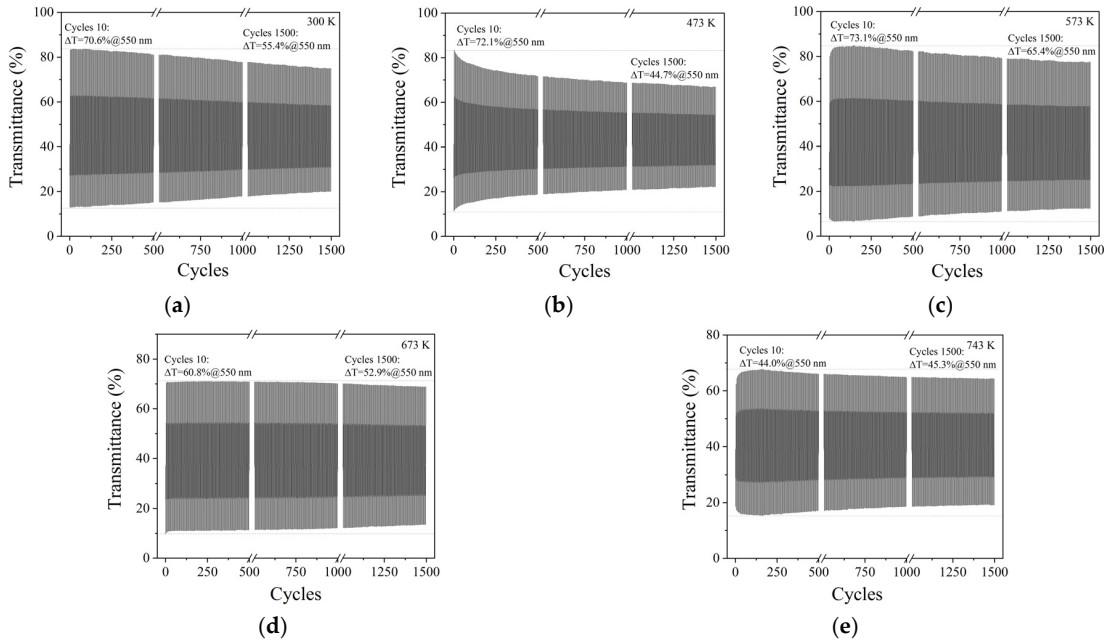

**Figure 8.** The transmistance curve of WO$_3$ film after 1500 cycles deposited at different substrate temperatures: (**a**) 300 K, (**b**) 473 K, (**c**) 573 K, (**d**) 673 K and (**e**) 743 K.

## 4. Conclusions

In this study, DC reactive magnetron sputtering method was used to deposit WO$_3$ film on the ITO substrate, and the microstructure and crystal structure were adjusted by changing the substrate temperature. The results show with substrate temperature increasing from 300 to 743 K, the WO$_3$ thin film changes from amorphous to monoclinic phase crystal state. Unexpectedly, when the substrate temperature is 573 K, the WO$_3$ films have a combined structure of amorphous and nanocrystalline. XPS measurement confirms that as the substrate temperature increases, the ratio of W(VI)/W(V) decreases, oxygen vacancies increase, and the binding energy of the thin film changes to a regular crystal structure. The electrochromic properties of the deposited WO$_3$ films were first improved and then degraded with the increase of substrate temperature. The WO$_3$ films deposited at substrate temperature of 573 K with Li$^+$ diffusion coefficient of $4.40 \times 10^{-11}$ cm$^2$/s, demonstrate the best optical modulation of 73.1% and

coloration efficiency of 49.2 cm$^2$/C at a wavelength of 550 nm. And the modulation amplitude retention rate of 89.5% after 1500 cycles is achieved, indicating the excellent cycle stability. These results indicate that changing the substrate temperature can effectively improve the electrochromic properties of the WO$_3$ film. This should attribute to the intermediate states between amorphous and nanocrystalline, which can provide more sites and short diffusion path for lithium ion injection/extraction.

**Author Contributions:** Conceptualization, H.-l.W.; methodology, Z.-j.X.; formal analysis, Q.S. and Z.-g.Z.; investigation, Y.-f.S. and H.-j.L.; writing—original draft preparation, Z.-j.X.; writing—review and editing, P.T.; supervision, M.-j.D. All authors have read and agreed to the published version of the manuscript.

**Funding:** This research was funded by Guangdong Special Support Program (No. 2019BT02C629), Scientific Research Fund of Guangdong Province (2016A030312015), GDAS'Project of Science Technology Development (No. 2018 GDASCX-0402), GDAS' Talent Introduction Project (No. 2020GDASYL-20200103110).

**Conflicts of Interest:** The authors declare no conflict of interest.

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
