# Peer review of "Enhanced Electrochromic Properties by Improvement of Crystallinity for Sputtered WO3 Film"

_coatings, doi:10.3390/coatings10060577_

Round 1

Reviewer 1 Report

Study entitled "Enhanced Electrochromic Properties by Improvement of Crystallinity for Sputtered WO3 Film" presents preparation of WO3 thin films consisting of different WO3 crystal structures. Thin WO3 films were grown under different temperatures on In2O3:Sn coated glass substrates by DC reaction magnetron sputtering technique using pure metallic tungsten as target. Crystallinity, chemical composition, oxidation state, microstructure, optical properties, electrochromic and electrochemical characterization and stability were subjected to study.

In the abstract and further in the text there is no clear explanation of in what way the published study is beneficial and important. The information provided is not convincing enough.

There are more (many) similar articles, is the solution published really such unique? What is a breakthrough? Represents this study a significant advance in the research area? If given from the impression one can get after reading the article, the answer is "no".

On the other hand, the article provides sufficiently carefully characterized materials. The experiments and comments are well conclusive, and the text presented is truly exemplary in this respect.

It is recommend to focus more on the comparison with previously published results. A possible solution to cope with this challenge is, for example, a table summarizing previous studies and results achieved so far, both in terms of layer composition with respect to various WO3 forms and stability, functionality, etc., all compared to the results of this study to its benefits were clearly obvious. The study would certainly be enriched by real photographs of prepared and tested samples. It is improved coloration efficiency of 42.9 cm2/C so crucial that it solves the existing problems? What value should be achieved?

The explanation given in passage: "It is calculated from the XPS element peak intensity that the atomic ratio W/O decreases with increasing substrate temperature, may be related to the oxidation state of the elements." (Page 3, Lines 96/97) is insufficient and the observed phenomenon needs to be clarified.

The use of notation such "W6+" or "W5+" is chemically incorrect as such record expresses discrete particles, which of course is not true. Please use chemically correct "W(VI)" or "W(V)".

Inserted SEM micrographs of film crosssection (Fig. 3 (a) - (e)) are completely illegible and therefore meaningless. Please enlarge them to the same size or cut out only a part so that it is not necessary to reduce it. Alternatively, the data can be presented in the form of Supplementary.

Please give more authentic explanation instead of statement "may be" in record "This may be due to the fact that the increased substrate temperature causes the agglomerate of particles and then grains coarse grow, resulting in a rougher surface roughness [25]." (Page 5, Line 137/138).

Excellent stability after 1500 cycles is discussed. But what is the time stability? Is the prepared material stable and does not change even over time?

There are a number of shortcomings in the manuscript with regard to word processing. Before next submission, a very careful revision of the entire text (tables and figures including) and removal of all transgressions and typographical shortcomings is recommended. The comments below do not claim to be complete! It only prove the validity of this allegation.

OTHER COMMENTS (Minor):
Page 1, Line 31: Please add missing space in record "electrochromic(EC) [1-3].".
Page 1, Line 44: Please add dot in record "Madhavi V et al" ("Madhavi V. et al is correct). Moreover as this is abbreviated "Et alii" the most correct form should be "Madhavi V et al"
Page 2, Line 53: Please add missing space in record "...In2O3:Sn(ITO with...".
Page 2, Line 53: Please check record "resistance of 8 Ω/square".
Page 2, Line 55: Again missing space "...as target(76.2 mm...". This error will no longer be commented on. Please check the whole manuscript thoroughly and correct whole text for such shortcomings (several times).
Page 2, Line 55: Please use correct "minus sign" instead of typographically incorrect "-" (hyphen) in record "8.0×10-4 Pa" and in the case of all other negative numbers in manuscript .
Page 2, Line 62: Please use correct symbol in the range in record "scanning range of 20o~80o". Symbol "~" is unacceptable. Please do not use hyphen "-" which is typographically also inappropriate.
Page 2, Line 62: Please use correct symbol for degrees (°) in record "scanning range of 20o~80o". It is unacceptable to use letter "o" in superscript. Please check the whole manuscript thoroughly and correct whole text for such shortcomings (several times).
Page 2, Line 68: Please use correct symbol in the range in record "300~800 nm". See comment "Page 2, Line 62" for details.
Page 2, Line 73: Please check record "-1 V". See comment "Page 2, Line 55" for details.
Page 2, Line 81: Please add space between number and unit in record "573K". Please check the whole manuscript thoroughly and correct whole text for such shortcomings (several times).
Page 2, Line 88: Why capital letter "C" is used in "...and Completeness monoclinic..." record?
Page 3, Fig. 1: Intensity units are missing in Fig. 1(a) and Fig. 1(b) on y-axis.
Page 3, Fig. 1: Please remove redundant dot "." between Fig. 1(a) and Fig. 1(b).
Page 3, Line 110: The record "530.7 and 532 eV" is incorrect with respect to an inconsistent number of significant digits. Correct record is "530.7 and 532.0 eV" or "531 and 532 eV". Nothing in between.
Page 4, Fig. 2(a)-(d): Please add missing space in axis labels "Intensity(a.u.)" and "Binging Energy(eV)" (correct should be "Intensity (a.u.)" and "Binging Energy (eV)"). Moreover "Binging Energy (eV)" is a nonsense - correct is "Binding Energy (eV)". Please correct.
Page 5, Line 134: Please add many missing spaces to record "(a)300 K;(b)473 K;(c)573 K(d)673 K and (e)743 K."
Page 5/6, Fig. 4(a)-(e): Please check corrupted label "Height Sensor ) μm" (?) and correct it.
Page 6, Fig. 4(f): Please change x-axis label "Substrate temperatures(K)" to "Substrate temperature (K)".
Page 6, Line 160: Please use correct greek letter for "v" in record (αhv)2.
Page 6, Line 164: Please correct record "...of the film, High substrate..." (unwanted capital letter H).
Page 6, Fig. 5(a): Please correct y-axis label "Tranmittance" ("Transmittance" is correct).
Page 7, Line 201: Please correct record "Figuer 7".
Page 8, Lines 206/207: Please avoid unwanted line breaks between number and units "-46.4 mC/cm2". Hard space should be used.
Page 8, Line 207: Variables (e.g. "tc and tb") should be written in italic. Please check the whole manuscript thoroughly and correct whole text for such shortcomings (several times).

Quite high similarity index (29 %) in Crossref was found.
Please rephrase:
Page 2, Lines 70-73; Page 3, Lines 93-94; Page 5, Lines 133-134; Page 6, Lines 146-147; Page 6, Lines 153-154; Page 6, Lines 155-157; Page 6, Lines 161-163; Page 7, Lines 170-172; Page 9, Lines 249-252; Page 9, Lines 254-258.

Author Response

Thanks for the reviewers’ comments, we have made substantial revisions on our manuscript.

When reviewer 2 responded to the file due to an inadvertent upload, no file was submitted. Therefore, the reviewer 1’s response and the reviewer 2 are merged into one document, I hope you can understand

The uploaded document is a reply to the reviewers, thank you very much for your valuable suggestions

Reviewer 2 Report

In this work, WO3 thin films with different crystal  structures were prepared by DC reactive magnetron sputtering method. The effects of substrate temperature on the structure, composition and electrochromic properties of WO3 films were studied.

The results show that the improvement of crystallinity for the deposited WO3 film was obtained with the increase of substrate temperature, which plays a major role in performance control.

The article is fairly structured.

 However, some issues must be fixed:

1)In fig.2 the authors should indicate a.u.as arb.units

2) In fig. 1 the authors should indicate arb.units as the unit of intensity

In fig. 1 the crystallographic directions are not readable.

3)The authors should use either ( ) or  / throughout the text  to indicate the unit of measurements consistently.

4)In fig. 3 the scale bars are not readable.

5)The inserts in Figs. 5 and 6 are not readable.

6)I suggest the authors to split Fig. 4 in more figures. Six figures in Fig. 4 are not readable.

7)The authors should explain the main novelty of their work in the introduction for readers not familiar with the topic.

8)Magnetron sputtering is an effective physical vapor deposition method which has been widely employed in the enterprise of thin film growth.

In the introduction the authors should refer to previous works about magnetron sputtering:

See for example:

[1]     Micro-Raman investigation of Ag/Graphene oxide/Au sandwich structure, Mater. Res. Express. (2019). https://doi.org/10.1088/2053-1591/ab11f8.

[2]     Structural and mechanical properties of amorphous AlMgB14 thin films deposited by DC magnetron sputtering on Si, Al2O3 and MgO substrates, Appl. Phys. A. 126 (2020). https://doi.org/10.1007/s00339-020-3316-z.

[3]     Graphene oxide on magnetron sputtered silver thin films for SERS and metamaterial applications, Appl. Surf. Sci. Volume 427 (2018) 927–933. https://doi.org/https://doi.org/10.1016/j.apsusc.2017.09.059.

 [4] Cu Thin Films Deposited by DC Magnetron Sputtering for Contact Surfaces on Electronic Components, Arch. Metall. Mater. 56 (2011) 903–908. https://doi.org/10.2478/v10172-011-0099-4.

Author Response

Thanks for the reviewers’ comments, we have made substantial revisions on our manuscript. The uploaded Word document returns the reviewer’s response, please consult.

Reviewer 3 Report

Qian Shi and co-workers investigate in their manuscript "Enhanced Electrochromic Properties by Improvement
of Crystallinity for Sputtered WO3 Film" the influence of process parameters on crystallinity, optical, and electrochromic properties of tungsten oxide thin films. A certain substrate temperature is identified to obtain optimized electrochromic properties. The used methods to characterize the samples are appropriate, the obtained results seem to be scientifically sound. The optimized values and their relation to the WO3 thin films crystallinity are for sure of interest for the readers of "Coatings". However, sometimes (details see below) it is quite hard to understand what the authors mean - a thorough grammar check should be done until the paper is ready for publication. Therefore, and because of the shortcomings (see the list below), I recommend that the paper might be published after minor revision.

Some more points:

1. line 22: "... of amorphous and nanocrystalline were obtained." - something is missing here.

2. line 33: "large range" instead of "large rang"?

3. line 66: "Bruker", not "Burker"

4. line 88: "... and Completeness monoclinic phase of WO3 ..." - something is missing.

5. line 92: This are no "XRD spectra"! The term "spectrum" stands for something which is energy-dependent. Here, the intensity is plotted vs. the diffraction angle, the energy of the diffracted X-rays is constant.

6. Figure 1: not all peaks are indexed.

7. line 95: "... is the external calibration source added during the XPS test." The authors should elaborate in more detail on that.

8. Figure 2: In all plots, "Binging" has to be replaced by "Binding".

9. Figure 3: Scale bars should be enhanced.

10. "Height Sensor" blocks the x-axis for all 2D AFM images". The axes labels for the 3D plots are too small. The 3D plots are superfluous, since all information is already visible in the 2D plots (or vice vesa).

11. Figure 5: "Transmittance", not "Tranmittance". Error bars for the inset in (b) are missing.

12. Line 163: The authors claim that reason for the increased band gap is "... because it refracts light and increases the band gap of the film," The authors should elaborate on that.

13. Figure 5(a): Transmittance and absorbance plot in (a) are related and one of the two plots can easily be removed.

14. Figure 5(b): Error bars are missing for the inset. Labels are way too small.

15. Line 188: "... which was superior to the reported before." Incomplete sentence. Reference should be added.

16. Figure 6: Error bars are missing in the inset.

17. Line 201: "Figure 7", not "Figuer 7"

18. Line 244: "Table 300"???

19. Line 271: "Appl. Optics", not "Appl. optics"

20. Line 307: "Surf Coat Technol", not "Surf Cota Tech"

21. A proper description of the AFM parameters (imaging mode, used tips) is missing.

Author Response

Thanks for the reviewers’ comments, we have made substantial revisions on our manuscript.The uploaded document is a reply to the reviewers, thank you very much for your valuable suggestions.

Reviewer 4 Report

In the manuscript, authors present the results of the structural, optical and electrical analysis of WO3 thin films deposited by DC reactive magnetron sputtering. They have discussed the influence of deposition temperature on WO3 properties and stated that by deposition temperature they can optimize electrochromic properties.

The manuscript is well structured. Content and obtained results are interesting and expressed in proper English language. However, the novelty presented in the manuscript is moderate. Searching the literature I found many publications related to electrochromic properties of WO3 deposited by magnetron sputtering at varied temperature. For example: Marszalek, K., 1989. Magnetron-sputtered WO3 films for electrochromic devices. Thin Solid Films 175, 227–233. https://doi.org/10.1016/0040-6090(89)90832-8; Chananonnawathorn, C., Pudwat, S., Horprathum, M., Eiamchai, P., Limnontakul, P., Salawan, C., Aiempanakit, K., 2012. Electrochromic Property Dependent on Oxygen Gas Flow Rate and Films Thickness of Sputtered WO3 Films. Procedia Engineering 32, 752–758. https://doi.org/10.1016/j.proeng.2012.02.008.

In the manuscript, there are several statements and conclusions that are not supported by experimental results, which should be clarified. For example, the authors stated that all samples have the same thickness of 400nm but cross-section SEM images and also UV-VIS transmittance data are not supporting this conclusion. Also, discussion and interpretation of UV-VIS transmittance data are not supported by measurements. Comparison of transmittance data at a single wavelength is not applicable in this case because of interference fringes presence. In this case, should be calculated and constructed mean transmittance for comparison purpose. Also, transmittance data (Fig 5a) are not consistent with Tauc plot data (Fig 5b). According to Fig 5a sample deposited at 473 K has the highest bandgap energy but according to Fig 5b, it doesn’t. This should be checked and clarified. Also, samples have double-layer structure (WO3 + ITO) and contribution of ITO was not discussed.

There are also other comments and suggestions (mostly minor) that can improve manuscript quality:

  • Row 54: … DC reaction magnetron … should be “reactive”
  • Row 59: “temperature gradient” What does it mean? Temperature is not constant during the deposition?
  • Row 78: “... shows thin films of WO3 films ...” sentence construction should be corrected.
  • Figure 1: not all diffraction peaks are indexed. Please add labels for them. 
  • Figurer 1: Subfigures should be properly labelled (a, b) and described in the figure caption. What is the difference between left and right subfigure? Only angular scale range? Why are missing data for the highest temperature in the right subfigure?
  • Figure 1: For sample deposited 673 K there is a weak diffraction peak slightly below 27 deg. but it’s not visible for sample deposited at 743K? The peak is not indexed? What is the origin of this peak and why it vanishes for 743 K sample?
  • Row 89: How is determined grain size? “ … grain size in the range of 12 nm …”? What does it mean? What is the min and max value of this range?
  • Figure 1: Is it preferred orientation presented in the WO3 layers?
  • Figure 1: X-axis label format is not consistent with the rest of the plots. The separator of units “/”
  • Figure 2: X-axis label: “binging energy”. Possibly should be “binding energy”.
  • Row 97: Statement “... may be related to the oxidation state of the elements. ...” is not clear. The oxidation state of which element?
  • Row 103: Not clear sentence: “Moreover, W4f7/2 and W4f 5/2 can be split into W6+ with higher binding energy and W5+ double peak with lower binding energy, W6+ also proves that the existence of WO3 chemical state. Figure 2c shows that the peak position of the W4f spin trajectory shifts to the”. I’m not expert for XPS, please elaborate on how it proves the existence of WO3 state if there is also W5+ state. W5+ is related to which state of WO3?
  • Row 107: Ratio of W6+/W5+ decrease with temperature? What is the explanation for this related to the previous discussion about WO3 state?
  • Row 111: “interstitial oxygen peak increased significantly, indicating that the oxygen vacancies of the film increased.” If you have more interstitial oxygen atoms, that does not mean necessarily that there are less lattice oxygen especially because of the increased ration of O/W?
  • Row 121 and Figure 3: In the text, it is stated that all films have the same thickness but this is not supported by cross-section SEM images. Also if you check transmittance data you can see that distance between adjacent maxima or minima is different from sample to sample indicating different thickness with the assumption of the identical index of refraction.
  • Figure 3: Please label the WO3 and ITO layers and substrate in cross/section images. Also, scale-bar in cross/section images is difficult to read.
  • Figure 4: In the AFM top-view surface images scale-bar is missing. Also in the 3D view, it is very difficult to read.
  • Figure 5. Transmittance and absorbance data are not consistent with the Tauc plot. Energy is inversely proportional to wavelength, so curve with transmittance drop at shortest wavelength should correspond to the curve at highest energy in Tauc plot but it does not?
  • Figure 8: Typing error in the caption: “Table 300”. It should be “temperature”

Author Response

(The authors gave the same response as above.)

Round 2

Reviewer 2 Report

I recommend the publication of this article. 

Author Response

Thanks to the reviewers for the revised comments on my article.

Reviewer 3 Report

Qian Shi and co-workers have improved their manuscript "Enhanced Electrochromic Properties by Improvement of Crystallinity for Sputtered WO3 Film" considerably. They sufficiently answered all my questions from my initial report. Therefore, I think that the paper should be published. Only some minor errors have to be corrected.

1. line 39: "... related to its the structure, ..." - remove the "the"

2. Line 88: "The results reveals that ..." - should be "The results reveal that ..."

3. Line 104: "There are continuous three spectrum corresponding to ..." - please reformulate.

4. Line 209: "... is appropriately increases to 573 K" - should be "... is appropriately increased to 573 K"

5. Line 328: "Surf. Coat. Technol.", not "Surf. Cota. Technol."

6. Line 337: "CrystEngComm", not "Crystengcomm"

Author Response

Thanks to the reviewers for their careful comments on my article. I have made corrections and the results are displayed in the uploaded file.

Reviewer 4 Report

The authors have taken the reviewers’ comments seriously and in the most part revised the manuscript accordingly. But in my opinion, there are still several conclusions and statements that are not supported by experimental results and before publications should be clarified:

Point 15: Row 121 and Figure 3: In the text, it is stated that all films have the same thickness but this is not supported by cross-section SEM images. Also if you check transmittance data you can see that distance between adjacent maxima or minima is different from sample to sample indicating different thickness with the assumption of the identical index of refraction.

Point 15: The experiment keeps the thickness of the WO3 film within a certain range of 300-400nm, while the thickness of the cross-sectional SEM image is basically satisfied. In the test, if the transmittance data is checked, the influence on the transmittance within this thickness range is very small.”

Please, label the WO3 and ITO layer in SEM cross-section images in Figure 3 to unambiguously proof thickness of WO3 layer, because as I disused in Point 15 SEM and transmittance data indicated significantly different WO3 layer thickness.

Point 18: Figure 5. Transmittance and absorbance data are not consistent with the Tauc plot. Energy is inversely proportional to wavelength, so curve with transmittance drop at shortest wavelength should correspond to the curve at highest energy in Tauc plot but it does not?

Point 18: Where absorbance coefficient ( and optical band gap ( satisfy the relationship as: , The greater the absorption rate, the smaller the . Experiments show that as the substrate temperature increases, the film transmission rate decreases, the absorption rate decreases, and its optical band gap increases, test results of compound experiments.

Conclusion “The greater the absorption rate, the smaller the ” is completely wrong. Energy gap Eg is related to transmittance sharp drop position (energy or wavelength in transmittance plot in Fig 5a) by optical energy gap definition. That means that by looking in Figure 5a it’s clear that sample deposited at 473K (red line) should have the highest value for Eg and samples deposited at 673K and 743K should have a lower value for Eg. To clarify that please add a plot of calculated absorption coefficient as a function of energy or wavelength.

Also in Rows 165-170 conclusion about transmittance drop is not supported by Fig 5a. Because transmittance variation (the result of interference) in visible range it’s not correct to make such conclusion only from looking at a specific wavelength. If you look at wavelength 700nm then sample 743K have the highest transmittance. It’s only correct to compare the calculated absorption coefficient which doesn’t show such oscillations.

Please specify also how the absorption coefficient used to construct Tauc plot was determined, and specially how the contribution of ITO layer was treated!

Also, please calculate thickness from interference minima and maxima position in Fig5a and that values compare to SEM cross-section images to support a conclusion about film thickness in range 300-400nm.

Author Response

Thanks to the reviewers for their valuable comments and careful correction of inappropriate things in my article. I have made corrections and the results are displayed in the uploaded file.

Round 3

Reviewer 4 Report

Authors have corrected the manuscript according to comments and suggestions.